# The TAM Subfamily of Receptor Tyrosine Kinases: The Early Years

**DOI:** 10.3390/ijms25063369

**Published:** 2024-03-16

**Authors:** Anne L. Prieto, Cary Lai

**Affiliations:** 1Department of Psychological and Brain Sciences, Indiana University, Bloomington, IN 47405, USA; aprieto@indiana.edu; 2Gill Center for Biomolecular Science, Indiana University, Bloomington, IN 47405, USA

**Keywords:** TAM, Tyro3, Axl, Mer, MerTK, tyrosine kinase, Gas6, protein S

## Abstract

The TAMs are a subfamily of receptor tyrosine kinases (RTKs) comprised of three members, Tyro3, Axl and Mer. Evidence in support of the existence of this subfamily emerged from a screen for novel RTKs performed in the laboratory of Dr. Greg Lemke in 1991. A PCR-based approach to selectively amplify tyrosine kinase-specific genes yielded 27 different tyrosine kinase genes, of which 13 were novel (the “*Tyros*”). Of these, *Tyro3*, *7* and *12* were more closely related to each other than to any other kinases and it was proposed that they constituted a novel subfamily of RTKs. Additional support for this hypothesis required determining the complete sequences for these receptor tyrosine kinases. By the end of 1991, full-length sequences for *Tyro7* (*Axl*) revealed a unique extracellular domain organization that included two immunoglobulin-like domains and two fibronectin type III repeats. In 1994, the complete sequences for *Tyro12* (*Mer*) and *Tyro3* were shown to have an extracellular region domain structure similar to that of Axl. In 1995, Gas6 and Pros1 were reported as ligands for Tyro3 and Axl, setting the stage for functional studies. The Lemke lab and its many trainees have since played leading roles in elucidating the physiological relevance of the TAMs.

## 1. The Cloning of the “Tyros”

It is a pleasure to participate in this volume devoted to the work of Dr. Greg Lemke, in whose lab I worked as a postdoctoral fellow from 1989 to 1992 at the Salk Institute. After joining the lab, I embarked on a project that would ultimately lead to the identification of multiple novel receptor protein tyrosine kinases, including a subfamily that we now call the “TAMs”. My colleague and co-author, Dr. Anne Prieto, has been working with receptor tyrosine kinases since 1996 and much of her efforts have focused on the TAM subfamily. The TAMs are known for their role as negative regulators of the innate immune system, as key mediators of the phagocytosis of apoptotic cells and as regulators of vascular integrity and permeability. They have also been identified as molecules that facilitate the entry of enveloped viruses, as enhancers of myelination and as receptors that promote the survival and metastasis of tumor cells [1,2,3,4,5]. 

When I joined the lab in 1989, Greg was a young assistant professor in the Molecular Neurobiology lab, where many groups were in pursuit of genes of interest in the nervous system. This was the golden age of cDNA cloning and the lab of Steve Heinemann, which was down the hall, was a few months away from identifying the first glutamate receptor, which they would report by the end of the year. The original goal of my project was to identify “interesting” genes expressed by Schwann cells, as at the time, Greg’s lab focused on the molecular biology of peripheral myelination. My graduate work had been devoted to working on genes that were selected for expression in a “brain-specific” manner and one issue to emerge from those efforts was that the selection criterion of brain specificity was much broader than we had originally envisioned.

Therefore, faced with another search for “interesting” genes, I thought that it might be better to focus on a particular class of molecules whose functional properties had been previously characterized, such as one known to be involved in some aspect of signal transduction. At the time, I had been excited by work in Drosophila, where it had been shown that the loss of function of a single receptor tyrosine kinase (“Sevenless”) led to a change in cell fate of a photoreceptor cell into a nonneuronal cone cell [6]. Although in retrospect, it seems that I was rather naïve, but I was hoping that we might find a tyrosine kinase that could play a “key role” in the Schwann cell lineage. 

In order to search for tyrosine kinases, we initially tried a hybridization-based library screening approach under the guidance of Rick Lindberg and Dave Middlemas in Tony Hunter’s lab at the Salk using degenerate oligonucleotide probes. Our interactions with the Hunter lab were facilitated by Gerry Weinmaster, a postdoctoral fellow in the Lemke lab who had also been a fellow in Tony’s lab. Rick and Tony had successfully used this approach to clone the receptor tyrosine kinase, “Eck” [7]. Although I had only attempted this method once, I observed a relatively high background in my library hybridizations and none of the isolated clones contained DNA sequences showing similarity to tyrosine kinases.

I then decided to use what at the time was still considered an “emerging technique”, PCR. In a paper published in March of 1989, Andrew Wilks used PCR to successfully identify two novel tyrosine kinase family members [8]. Andrew had used degenerate primers corresponding to the highly conserved regions within the tyrosine kinase catalytic domain. I elected to use a modified version of his strategy with primers that were both longer and fully degenerate. The motivation for these changes was that I thought that they might enable us to capture a higher percentage of the tyrosine kinases in our cDNA samples. However, one concern regarding the use of highly degenerate primers (4000- and 2000-fold, respectively, for the upstream and downstream primers) was that it could negatively affect the extent of hybridization during the annealing step of the PCR. I, therefore, decided to use a very long annealing period (“programmed at 5 min” but actually closer to 8 min with our primitive thermal cycler that used water to both heat and cool the samples). The annealing step was performed at a low temperature (37 °C) and with a relatively high concentration of primers. 

In order to focus on Schwann cell-specific expression, I prepared cDNA from RNA obtained from the sciatic nerves of neonatal postnatal (P) day 2/P3 rats. The first strand cDNAs were used in a series of subtractive hybridization procedures designed to enrich for the Schwann cell component of gene expression. I had previously gained considerable experience with nucleic acid hybridization techniques from working in the laboratory of Eric Davidson while at Caltech in 1977. In addition, I was familiar with the work of a former roommate, Mark Davis, who had used subtractive procedures to facilitate the cloning of the T cell receptor [9]. In retrospect, a major benefit of using the subtractive procedures for our project was to reduce the “effective concentration” of the most abundant RTKs. We were fortunate as the pilot PCR amplification in August of 1989 appeared promising as it yielded a broad but readily detectable band at around 200–230 bp, which was within the predicted size range. We performed a second amplification and it was these products that were used for the subcloning. The DNA sequences of individual clones were determined and of the first 50 to be analyzed, 44 bore amino acid sequences conserved in tyrosine kinases. There were 22 different genes in this initial set and 11 of the 22 appeared to be novel tyrosine kinases.

When completed (155 RTK-like sequences from 168 isolates), our search yielded a total of 27 tyrosine kinase genes, of which 13 were novel [10]. Of these thirteen, eleven showed similarity to receptor-type tyrosine kinases while two appeared to be of the non-receptor class. As shown in Figure 1, these sequences were clustered into subfamilies based on amino acid sequence similarity. Anne Marie Quinn, a bioinformatic specialist at the Salk, played a leading role in generating the sequence comparisons. We named the novel genes the “*Tyros*”. Five of the novel RTKs were grouped into the “Eph” subfamily, with *Tyro1*, *Tyro4*, *Tyro5*, *Tyro6* and *Tyro11* now known as *EphA4*, *EphA3*, *EphB2*, *EphB3* and *EphB4*, respectively. In addition to expanding the number of known Eph subfamily members, our screen identified *Tyro2* as a novel member of the *EGFR* subfamily (ErbB4) and *Tyro9* as a new member of the FGF receptor subfamily (FGFR4). Therefore, although we did not know it at the time, our project had completed the roster of *EGFR* and *FGFR* subfamily members. One gene (*Tyro10*) became the focus of our early interest as it was related to *Trk*, which had been reported as a novel receptor for nerve growth factor (NGF) a few weeks prior to the publication of the Tyros [11]. Ultimately, *Tyro10* did not prove to be a novel receptor for the neurotrophins, but was identified as a receptor for collagen by a group at Regeneron Pharmaceuticals [12] and by Tony Pawson’s group [13]. *Tyro10* is now known as the “Discoidin domain tyrosine kinase receptor 2” (DDR2).

The remaining three novel RTKs (*Tyro3*, *Tyro7* and *Tyro12*) showed some similarity to the insulin receptor subfamily but they were more closely related to each other than to any previously described RTK. We decided to group these as a novel subfamily of its own. These now go by the names of Tyro3, Axl and Mer, respectively, and together form the “TAM” subfamily of receptor tyrosine kinases. A comparison of all known tyrosine kinases performed in 2002 shows that the TAMs are indeed more closely related to each other than to any other human tyrosine kinases (see Figure 2, modified from the Kinome Poster made by Cell Signaling Technologies) [14].

Based on Northern blot analyses, *Tyro3*, *Axl* and *MerTK* were all detected in Schwann cells grown in vitro in the absence of forskolin, with Axl being expressed at relatively high levels. The TAMs were also detected in the postnatal brain, but of these, *Tyro3* was expressed at significantly higher levels. Using hybridization in situ with the assistance of Alan Watts and Graciela Sanchez-Watts who were in the lab of Larry Swanson, we showed that *Tyro3* had readily detectable levels of expression in the cortex and prominent expression in the CA1 region of the hippocampus relative to that observed in the CA3 (see Figure 3, reprinted from [10]). The high levels of expression in these brain regions important for cognitive function made *Tyro3* one of our top candidates for additional analysis. 

The cloning of the Tyros represented an embarrassment of riches with too many choices to consider for future analyses. Our original goal had been to identify genes that were specifically expressed by Schwann cells, but it appeared that at least 10 of the 11 novel RTKs did not satisfy this criterion as they were all detected in brain samples in our Northern blotting assays. One exception was *Tyro9*/*FGFR4*, where signals were very low, precluding an accurate assessment of its cell type-specific expression. Accordingly, our future efforts largely focused on the roles of these molecules in the central nervous system. Although the Lemke lab has since worked on multiple RTKs to emerge from this screen and has made significant contributions to our understanding of the role of ErbB4 (for example, [15]) and the role of EphA receptor signaling in the developing nervous system (reviewed in [16]), as the years went by, the TAM subfamily became a major focus of the lab’s efforts.

It is important to note that despite the “novel” status of many of the Tyros at the time of publication in May of 1991 [10], many groups from around the world were identifying novel RTKs. For example, Juha Partanen and the group of Kari Alitalo published the complete sequence of FGFR4 in June of 1991 [17] and they had published a partial sequence of that gene at the time of the Tyro paper submission in November of 1990 [18].

Although our studies had strongly suggested that *Tyro3*, *Tyro7* and *Tyro12* together comprised a novel subfamily of receptor tyrosine kinases, it is important to note that we had identified only small regions of the genes encoding these receptors. The demonstration that these indeed represented a new subfamily would require obtaining the complete coding regions for each of the receptors. A summary of these cloning efforts can be found in Table 1.

## 2. The Cloning of Axl

Multiple groups reported the cloning of *Tyro7*-like homologs in 1991 within a few months of our publication of the Tyros. Each of the groups recognized that it represented a novel receptor tyrosine kinase based on its unique extracellular domain composition that consisted of a combination of two immunoglobulin (Ig)-like domains and two fibronectin (FN) type-III repeats.

Axl was first studied as “gene X”, by Edison Liu while he was working in Michael Bishop’s lab [33]. They were interested in identifying genes that were involved in the changes that drove the progression of chronic myelogenous leukemia (CML) from an initial chronic phase into a subsequent blast crisis stage. At the time, the fusion of the BCR and ABL genes on chromosomes 9 and 22, was known to occur in a majority of CML patients. This event was thought to be responsible for the chronic phase of the disease. In contrast, the transition of CML into the blast crisis stage was poorly understood. DNA isolated from peripheral blood leukocytes and bone marrow obtained from blast patients was used in gene transfer and tumorigenesis assays to identify candidate genes. A series of *RAS* and *RAF* mutants with transforming potential were identified and, in addition, another unknown gene that had lower transforming potential was also isolated. This unknown gene (“gene X”) was independently isolated from two different patients, one in the chronic phase CML and another in the blast crisis stage. DNA prepared from these two patients was capable of inducing tumors in nude mice. 

In October of 1991, Edison Liu’s lab published the characterization of gene X [27]. They accomplished this by first isolating cell lines from secondary tumors and then preparing cosmid libraries to search for clones that contained human DNA. Using exonic regions from these genomic clones, they were able to isolate full-length cDNA clones whose sequence predicted a novel receptor protein tyrosine kinase, which they named AXL for “anexelekto” (uncontrolled, in Greek). The *AXL* gene encoded a polypeptide of approximately 140 kDa. It contained an extracellular domain with two sequential Ig-like domains followed by two FN type-III repeats. As this domain combination had not been previously identified in other RTKs, Axl was thought to define a novel subfamily of RTKs. A month later, Janssen et al. independently identified a gene with transforming potential by transfecting NIH-3T3 cells with DNA also isolated from a CML patient [28]. They named this gene, “*UFO*”, for “unidentified function”. Like *AXL*, it shared the same extracellular region domain structure. The first full-length murine homolog, *Ark*, was published in October of 1991, using a screen with the *Bek*/FGFR2 tyrosine kinase domain [26]. Because of the presence of the Ig-like and FN-type III domain structure, they named it Ark, for “*a*dhesion-*r*elated *k*inase”. Ark is the murine homolog of Axl.

## 3. The Cloning of Mer (MERTK)

In 1992, a publication from the Hanafusa laboratory at Rockefeller University described the characterization of a novel oncogene derived from the avian retrovirus, RPL30 [34]. This viral oncogene, which they named v-eyk, encoded a fusion protein that included a portion of the viral envelope protein gp37 and a region thought to encode a functional tyrosine kinase domain (p69^gp37-Eyk^). This fusion protein was necessary and sufficient for the transforming properties of the RLP30 virus. v-Eyk was able to transform chicken embryo fibroblasts and exhibited constitutive tyrosine kinase activity as evidenced by the elevation of tyrosine-phosphorylated proteins in immunoprecipitates from v-eyk transformed cells. Initially, they named the viral oncogene v-ryk but subsequently renamed it v-eyk [29] as the name “ryk” had been used for another gene [35]. In 1994, the Hanafusa group described the cloning of c-*EYK* from chicken [29]. The sequence revealed that it encoded a receptor tyrosine kinase having two extracellular Ig-like domains and two FN type-III repeats as observed in AXL/UFO/Ark. In contrast, the viral oncogene lacked both the extracellular and transmembrane sequences. The c-*EYK* sequence and tissue-specific pattern of expression led the authors to conclude that c-*EYK* isolated from chicken did not correspond to the avian homolog of *AXL* but instead represented a novel RTK. In 1994, the laboratories of Shelton Earp and Ralph Snodgrass were able to isolate a full-length clone corresponding to the human homolog of c-*EYK* through expression screening using anti-phosphotyrosine antibodies [31]. They designated this clone as “c-*MER*”, a term that reflected its expression in **m**onocytes and tissues of **e**pithelial and **r**eproductive origin. Like AXL/UFO/Ark, the human c-EYK/c-MERTK extracellular region contained two Ig-like domains and two FN type-III repeats. The intracellular domain had 83% similarity and 71% identity to v-eyk. This novel tyrosine kinase was not expressed in normal B- and T-lymphocytes but, unlike AXL was expressed in numerous neoplastic B- and T-cell lines. The isolation and characterization of murine c-*Mer* were reported by this group in the following year [30]. In 1995, Ling and Kung also reported the isolation of a human homolog of c-*MERTK* from a human glioblastoma tumor, which they designated *NYK* for “N-CAM related tyrosine kinase” [32], based on the domain organization often found in neural cell adhesion molecules [36].

## 4. The Cloning of Tyro3

The Lemke lab reported the sequence of the full-length murine *Tyro3* clone in 1994 [37], but by that time, multiple groups had already published the complete (or near-complete) sequences of both human and murine homologs of this receptor (see Table 1). In contrast to the oncogene search-based approaches used to find *AXL* and *MERTK*, the groups that identified *Tyro3* homologs used PCR and hybridization-based strategies. In March of 1994, a human *Tyro3* homolog named “*SKY*” for sea-related protein kinase was identified by low-stringency screening of a human HepG2 hepatoma cDNA library using a region encoding the tyrosine kinase domain of chicken “c-*SEA*” [24]. In April of 1994, Paul Godowski’s group at Genentech used a degenerate PCR-based strategy to isolate “*RSE*” (for receptor sectatoris) [23] from human brain cDNA. They used the human cDNA to also obtain the full-length murine clone. These clones encoded receptor protein tyrosine kinases having the same extracellular domain structure as AXL.

In addition, in March of 1994, a murine *Tyro3* homolog was identified using a PCR-based strategy from the fetal mouse brain (named “*Brt*” for brain tyrosine kinase) [19]. This clone contained sequences at both the N-and C-terminal regions that differed from our sequence and the murine sequence from Genentech. In the same month, another group identified a human Tyro3 homolog named “*tif*”, although their clone had a predicted N-terminal methionine residue located within the second Ig-like domain as well as an altered C-terminus [25]. And in back-to-back papers from the laboratory of Kathryn and Philip Crosier in New Zealand [20,21], full-length sequences of both murine and human homologs of Tyro3, were published under the name of “*Dtk*”. In 1996, a clone was isolated by Patricia Maness’ group from a chicken embryo retina library, which they designated “*REK*”. This receptor bears the same domain extracellular domain organization as the TAMs, with the closest homology to *Tyro3* [22].

In order to obtain a full-length murine *Tyro3* clone for the Lemke lab, we screened a mouse brain cDNA library with the original PCR-derived rat *Tyro3* clone. Our efforts were greatly facilitated by Jim Boulter, an expert on phage-based library construction and screening. Jim worked in the laboratory of Steve Heinemann and it was his molecular expertise that facilitated the cloning of the first glutamate receptor. The *Tyro3* cDNA clone that we obtained, “18A”, served as the starting point for expression studies in the Lemke lab and it is still in use today as the murine homolog of this receptor. The first expression study was performed by Martin Gore, who generated a stable Tyro3 cell line using Rat2 cells. Martin used these to demonstrate that Tyro3 conferred the ability of the cells to grow in soft agar, a property not shared by normal Rat2 cells [37]. This was the first demonstration that Tyro3 could potentially serve in an oncogenic role. As part of these efforts, the first rabbit polyclonal anti-Tyro3 antibody (5424) was produced using GST fusion proteins that contained the C-terminal 107 amino acids of this receptor. This has proven to be useful for both Western blotting and immunoprecipitation. Therefore, by the end of 1994, three human and three murine cDNA clones containing the entire coding region of Tyro3 had been reported.

## 5. The Identification of the TAM Subfamily Ligands

By the end of 1994, Tyro3, Axl and Mer had been recognized as a novel subfamily of RTKs. To meaningfully study these receptors, it was important to identify their cognate ligands. The TAMs are activated by two well-characterized ligands, protein S (Pros1) and growth arrest-specific gene 6 (Gas6). Three additional ligands that promote Mer-mediated phagocytosis (Tubby, tubby-like protein and Galectin 3) have been reported [38,39]. Additional characterization of the interaction of these ligands with the TAMs will be required to understand their relationship to this receptor family. Gas6 and Pros1 exhibit extensive amino acid sequence similarity and share a common domain organization [40]. In the amino-terminal portion, they contain a “Gla domain” that is characterized by a series of γ-carboxyglutamic acid (gla) residues that are capable of binding to negatively charged phospholipids in a Ca^+2^-dependent manner. In Pros1, the Gla domain is followed by a loop region that contains a thrombin cleavage site [41,42,43]. This loop region is not present in Gas6. These domains are followed by four epidermal growth factor (EGF)-like repeats and two globular (G1 and G2) domains that are related to those found in the steroid hormone-binding globulin (SHBG). These G domains can bind to the TAMs, an event that leads to receptor dimerization and activation. In addition to this interaction, the binding of the Gla domain to the phospholipid, phosphatidylserine, (PtdSer) is also important for TAM activation [44]. When γ-carboxylated, the ligands are able to simultaneously engage both PtdSer and a TAM. Although typically confined to the inner leaflet of the plasma membrane, PtdSer relocates to the outer leaflet and serves as a marker of apoptotic cells. The optimal activation of the TAMs requires a tripartite interaction of PtdSer, Gas6/Pros1 and the TAMs [45,46].

The identification of TAM ligands was first reported in February 1995. First, a group from Amgen described the isolation and identification of Gas6 as an activating ligand for Axl [47]. At about the same time, in a collaborative effort between the Lemke lab and a team at Regeneron Pharmaceuticals [48], Gas6 was identified as an activating ligand for Axl while Pros1 was identified as an activating ligand for Tyro3/Sky. The identification of Pros1 as a ligand was unexpected as it had been long known as an abundant serum protein involved in blood coagulation [41,42]. Later that year, Ohashi et al. [49] and Godowski et al. [50] were able to demonstrate the activation of Tyro3/SKY/RSE by Gas6. The Godowski group subsequently showed that a truncated form of Gas6, composed of only the G-domain, was capable of binding to both Axl and Tyro3/RSE [51]. That same year, Nagata et al. [52] demonstrated that Gas6 could also bind and activate Mer, thus, by 1996, it was identified as a common ligand for all 3 TAMs. These studies were extended by Nyberg et al. [53], who demonstrated that although Pros1 could engage Sky/Tyro3 through the G domains, activation of this receptor was significantly diminished in the absence or blockade of the Gla domain. It should be noted that the relative contribution of the activating properties of these two ligands for each of the TAM receptors was not elucidated for nearly two decades. In 2014, a publication from the Lemke lab played a key role in improving our understanding of how these ligands engage each of the TAM receptors [45]. It is now known that Gas6 activates all three of the TAM receptors, while Pros1 activates Tyro3 and Mer, but not Axl. These studies also reaffirmed the importance of the Gla domain in receptor activation. 

## 6. The Lemke Lab Has Played a Major Role in Elucidating the Biological Relevance of the TAMs

Since its inception, the contributions of the Lemke lab to the field of TAM biology have been extensive [10,37,45,48,54,55,56,57,58,59,60,61,62,63,64,65,66,67]. The phenotypic analysis of TAM receptor knockout mice provided early insights into the functional relevance of these receptors. One early finding was the role of the TAMs in mammalian spermatogenesis [61]. The three TAM receptors are expressed in the Sertoli cells, which provide trophic support for the germ cells. In the absence of all the TAMs, this trophic support is lost resulting in the progressive death of germ cells and impaired spermatogenesis. Work from the lab also defined the roles of the TAMs in the regulation of innate immunity through their interaction with cytokine signaling pathways [66] and contributed to the establishment of these receptors as potent immunosuppressors [5,68]. 

The Lemke lab has also made important contributions to the field of neuroimmunology. During the developing nervous system, there is large-scale cell death by apoptosis. This programmed cell death is also observed in zones that remain neurogenic in the postnatal and adult mice. In 2016 [57], they were able to demonstrate the TAM-mediated microglial phagocytosis of apoptotic cells in two neurogenic regions, the subgranular zone of the hippocampal dentate gyrus (DG) and the subventricular zone. They also determined that TAM-deficient microglia exhibit deficits in migration and process motility under conditions of damage and inflammation. 

In studies focusing on microglial function in an animal model of Alzheimer’s disease (APP/PS1), ref. [60] the Lemke lab showed that the loss of either Mer or Gas6 exacerbated the number of animals that suffered early sudden death resulting from seizures. In these APP/PS1 mice, the seizures occur at a young age and are believed to be driven by activity in the dentate gyrus. The authors correlated the increased epileptogenic activity in the Mer-deficient animals with an increase in synaptic number between hippocampal DG and CA3 neurons. These findings led the authors to conclude that the increased mortality was due to an increase in excitability caused by a reduction in synaptic pruning in the TAM deficient.

Recent work has defined a role for the TAM receptors and their ligands in microglia associated with amyloid plaque [59]. This work is remarkable in that it demonstrates that although the phagocytosis of amyloid plaque is impaired in animals with TAM-deficient microglia, contrary to expectation, these mice have a lower burden of dense-core amyloid β plaque. These findings implicate TAM-dependent microglial phagocytosis in the formation of dense-core amyloid β plaque deposits. The authors have proposed a model in which the dense-core plaques are “constructed” via the microglial phagocytotic processing of loosely organized amyloid fibrils. The authors have argued that the formation of the dense-core plaques may have a protective effect and draw an analogy to the formation of “granulomas” in tuberculosis [59,69]. These recent studies implicate the TAM receptors and their ligands in Alzheimer’s disease and raise issues concerning the development of therapeutic strategies that target amyloid plaque. These are a few examples highlighting the significance and breadth of the Lemke lab’s contributions to TAM receptor biology.

In conclusion, although our PCR-based search did not achieve our initial goal of identifying Schwann cell-specific genes, the strategy of focusing on a known family of signaling molecules provided an alternative pathway to success. The discovery of the Tyros opened many avenues of pursuit for the Lemke lab and the TAM subfamily became a major focus of its efforts. The work has continued in the labs of Greg’s many trainees, whose accomplishments are featured in this special issue. 

## Figures and Tables

**Figure 1 ijms-25-03369-f001:**
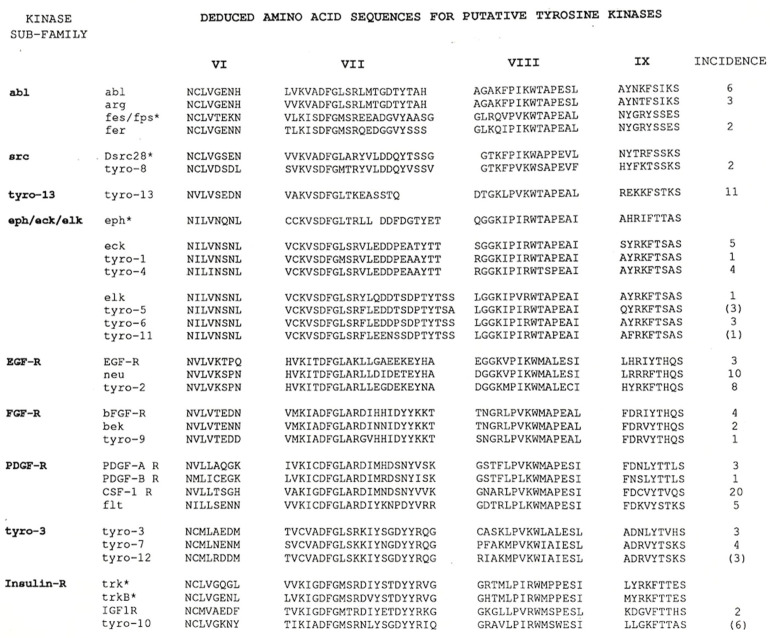
Deduced amino acid sequences of PCR product subclones. This figure, which has been reproduced from Lai and Lemke, 1991 [10], shows the list of 27 different tyrosine kinase genes identified in the PCR-based screen. Those marked with an asterisk (*) were not cloned in this project but have been included to facilitate the sequence comparisons. Based on the alignment of these short regions of amino acid sequence, *Tyro3*, *Tyro7* and *Tyro12* were more closely related to each other than to other identified RTKs and have been designated as a separate subfamily. For a more detailed description, please refer to the complete figure legend in Lai and Lemke, 1991. Reprinted with permission from Ref. [10]. Copyright 1991, Elsevier.

**Figure 2 ijms-25-03369-f002:**
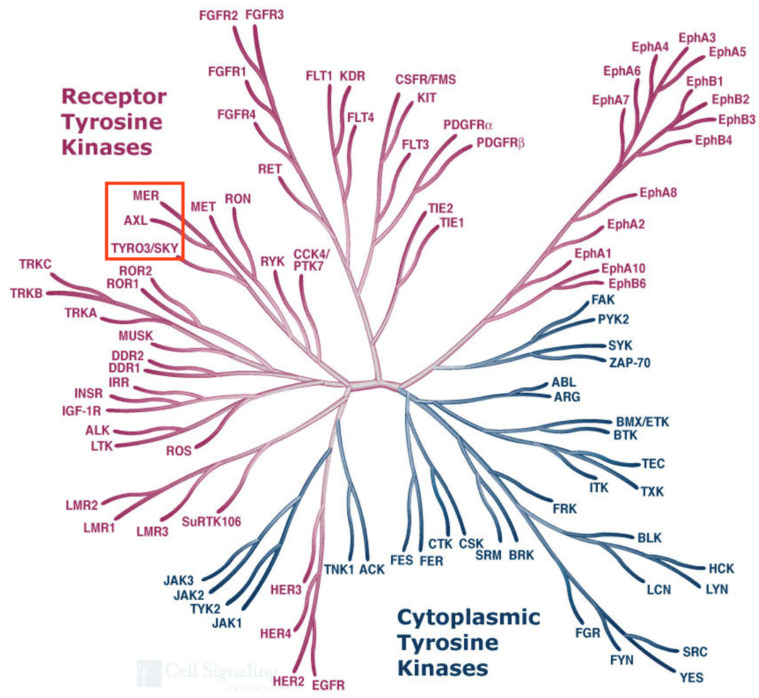
Dendrogram of the human tyrosine kinases. This image is a modified version taken from the Human Kinome poster associated with Manning et al., 2002 [14], that shows the relatedness within the tyrosine kinase subset of protein kinases. The full-length sequences of human Tyro3, Axl and Mer are more closely related to each other than they are to any other RTKs. Illustration reproduced courtesy of Cell Signaling Technology, Inc., Danvers, MA, USA www.cellsignaling.com, accessed on 22 December 2023).

**Figure 3 ijms-25-03369-f003:**
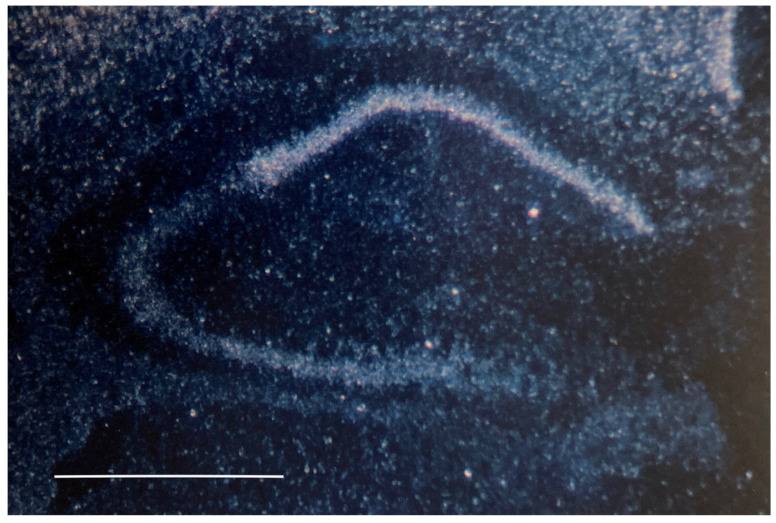
Tyro3 expression in the adult rat hippocampus. In situ hybridization of Tyro3 reveals a prominent signal in the CA1 field of rat hippocampus with a weaker signal in the CA3 field. This is a cropped version of panel 3a in Figure 5 from [10]. The scale bar represents 1 mm. Reprinted with permission from Ref. [10]. Copyright 1991, Elsevier.

**Table 1 ijms-25-03369-t001:** Tyro3/Axl/Mer RTK subfamily. This table summarizes the names assigned to the different cDNA clones isolated by the first groups to work in the TAM field.

Subfamily	Human	Rodent	Avian
Tyro3	*RSE*, *SKY*, *DTK*, *TIF*	*Tyro3*, *Dtk*, *Brt*	*REK*
Axl	*AXL*, *UFO*	*Tyro7*, *Ark*	
Mer	*MERTK*, *NYK*	*Tyro12*, *Mertk*	*EYK*

**Tyro3:** *Brt*, for brain tyrosine kinase, Fujimoto, J. and Yamamoto, T., 1994 [19]; *DTK*, for development tyrosine kinase, Crosier K.E., Hall, L.R., Lewis, P.M., Morris, C.M., Wood, C.R., Morris, J.C., and Crosier, P.S., 1994 [20]; *Dtk*, Crosier, P.S., Lewis, P.M., Hall, L.R., Vitas, M.R., Morris, C.M., Beier, D.R., Wood, C.R., and Crosier, K.E., 1994 [21]; *REK*, for retina-expressed kinase, Biscardi, J.S., Denhez, F., Buehler, G.F., Chesnutt, D.A., Baragona, S.C., O’Bryan, J.P., Der, C.J., Fiordalisi, J.J., Fults, D.W., and Maness, P.F., 1996 [22]; *RSE,* for receptor sectatoris, Mark, M.R., Scadden, D.T., Wang, Z., Gu, Q., Goddard, A., and Godowski, P.J., 1994 [23]; *SKY,* for sea-related protein kinase, Ohashi, K., Mizuno, K., Kuma, K., Miyata, T., and Nakamura, T., 1994 [24]; *TIF*, for tyrosine kinase with immunoglobulin-like and fibronectin III structures, Dai, W., Pan, H., Hassanain, H., Gupta, S.L., and Murphy, M.J Jr., 1994 [25]; *Tyro3*, for tyrosine kinase-3, Lai, C. and Lemke, G., 1991 [10]. **Axl**: *Ark,* for adhesion-related kinase, Rescigno, J., Mansukhani, A. and Basilico, C., 1991 [26]; *AXL,* for anexelekto, O’Bryan, J.P., Frye, R.A., Cogswell, P.C., Neubauer, A., Kitch, B., Prokop, C., Espinosa, R. III, Le-Beau, M.M., Earp, H.S and Liu, E.T., 1991 [27]; *Tyro7*, for tyrosine kinase-7, Lai, C. and Lemke, G. 1991 [10]; *UFO,* for unidentified function, *Janssen*, J.W., Schulz, A.S., *Steenvoorden*, *A.C.*, Schmidberger, *M.*, Strehl, *S.*, Ambros, P.F., and *Bartram*, *C.R.*, 1991 [28]. **Mer**: *EYK,* for East Lansing tyrosine kinase, Jia, R. and Hanafusa, H., 1994 [29]; c-*Mer*, for its expression in monocytes and tissues of epithelial and reproductive origin, Graham, D.K., Bowman, G.W., Dawson, T.L., Stanford, W.L., Earp, H.S., and Snodgrass, H.R., 1995 [30]; *MER*, Graham, D.K., Dawson, T.L., Mullaney, D.L., Snodgrass, H.R. and Earp, H.S., 1994 [31]; *NYK,* for N-CAM-related tyrosine kinase, Ling, L. and Kung, H.J., 1995 [32]; *Tyro12*, for tyrosine kinase-12, Lai, C. and Lemke, G. 1991 [10].

## Data Availability

No new data were created or analyzed in this study. Data sharing is not applicable to this article.

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
