# Peer review of "The TAM Subfamily of Receptor Tyrosine Kinases: The Early Years"

_ijms, 2024, doi:10.3390/ijms25063369_

Round 1
Reviewer 1 Report
Comments and Suggestions for Authors
See pdf

Author Response
Comments for Reviewer #1
- The reviewer considered that “RTK” (receptor tyrosine kinase) was the generally more accepted abbreviation for “RPTK” (receptor protein tyrosine kinase) and we have replaced RPTK with RTK throughout the manuscript. Please note that this change also includes the title.
- We have accepted all changes to the text requested by Reviewer 2. The “duplicate” figure legends appear to have arisen as a formatting issue from the journal site and we have retained the figure titles in the body of the manuscript and have only inserted the legends at the end of the text.
- We have now highlighted the TAM family members Tyro3, Axl and Mer in the dendrogram shown in Figure 2 to better draw the eye of the reader to this family of RTKs.
- We have modified the Figure legend for Table 1 to list all authors on the citations as requested. Now the senior authors names are visible.
- We do not know why Tyro3 is the official gene name. On the Human Genome Organization Gene Nomenclature Committee site, the previous symbol is shown as “Rse” and the currently approved name is “Tyro3”. The first group to publish the correct full-length human Tyro3 cDNA used the name “Sky” and their work preceded that of the group at Genentech (Rse) by a few weeks, therefore the gene name was not awarded to groups that published full-length sequences prior to the Lemke lab, which was in 1994.
Reviewer 2 Report
Comments and Suggestions for Authors
This article about the TAM family receptors is a unique look at the discovery of new tyrosine kinase receptors since the original finding of the EGF/FGF/PDGF receptors. The authors have done a good job of acknowledging the many individuals who contributed to the identification of the TAM receptors. In this way, numerous individuals were given credit for prior work that led up to the cloning of these unknown receptors, including the methodology used. A historical record of this information is important, as this is rarely recognized by the current generation of investigators.
The impact of this article will be strengthened if the biological outcomes of these receptors were cited. For example, TAM receptors function in microglia as sensors for the phagocytosis of dying brain neurons in amyloid plaques found in Alzheimer’s disease. Surprisingly, some amyloid plaques are protective and not damaging (Huang et al 2021). This unexpected conclusion indicates current strategies to remove dense core plaques in Alzheimer’s disease may do more harm than good. In addition, TAM receptors are recognized as essential outside the nervous system--in spermatogenesis and innate immunity. A list of key references should be provided to illustrate the impact of TAM receptors in biology.
Minor issues
1. The nomenclature is confusing (TAM, Tyro, Axl, Ark, Brt, Sky etc), not even to mention the numeration. Though it is unlikely a new naming scheme will be adopted, Table 2 should include the definition of the acronyms.
2. More explanation of the ligands for these receptors (protein 5, Gas6 etc) needs to be included, since their identification is important in defining the function of the receptors.
Articles which should be cited
Lu Q, Gore M et al (1999) Tyro-3 family receptors are essential regulators of mammalian spermatogenesis. Nature 398, 723-728.
Rothlin CV, Ghosh et al (2007) TAM receptors are pleiotropic inhibitors of the innate immune response. Cell 131, 1124-1136
Fourgeaud L, Traves PG et al (2016) TAM receptors regulate multiple features of microglia physiology. Nature 532, 240--244.
Huang Y, Happonen KE, Burrola PG, O’Connor C, Hah N, Huang L, Nimmerjahn A, Lemke G (2021) Microglia use TAM receptors to detect and engulf amyloid plaques. Nature Immunology 22: 586-594.
Author Response
Comments for Reviewer #2
- The Reviewer suggested that the impact of the article would be strengthened with the inclusion of some biological outcomes mediated by the TAMs. Specifically, the reviewer mentions work on the phagocytosis of apoptotic neurons and the potential role of the TAMs in Alzheimer’s disease described in Huang et al 2021. We have addressed this by doing 2 things: we introduced a very brief summary of different biological effects of the TAMs in the first paragraph of the document, and we also introduced another section at the end of the document highlighting Dr. Lemke’s contributions, among them the work on neuronal apoptosis and also his work on Alzheimer’s (Huang et al 2021). This new paragraph also incorporates the 4 requested references with brief descriptions of the work within. Because the emphasis of our article was on the “early days” of the TAMs and the requested work was beyond the time format that we had originally conceived, we deemed it more appropriate to highlight Dr Lemke’s work on the TAM receptors and its ligands after the early 1990’s period as a separate paragraph and in the hope to honor the reviewer’s request.
- Minor issue #1: We agree with the reviewer that the nomenclature is confusing. We have added the definition of the acronyms to the Table 1 legend.
- Minor issue #2: We have now expanded the section on the ligands, Gas6 and Protein S, but have largely limited our scope to the early years, with a brief statement concerning the current view of how these ligands engage the TAM receptors.